# Newborn Screening for Severe Combined Immunodeficiency: Do Preterm Infants Require Special Consideration?

**DOI:** 10.3390/ijns7030040

**Published:** 2021-07-08

**Authors:** Anne E. Atkins, Michael F. Cogley, Mei W. Baker

**Affiliations:** 1Wisconsin State Laboratory of Hygiene, University of Wisconsin School of Medicine and Public Health, Madison, WI 53706, USA; anne-atkins@uiowa.edu (A.E.A.); michael.cogley@slh.wisc.edu (M.F.C.); 2Genetics and Metabolism Division, Department of Pediatrics, University of Wisconsin School of Medicine and Public Health, Madison, WI 53706, USA; 3Center for Human Genomics and Precision Medicine, University of Wisconsin School of Medicine and Public Health, Madison, WI 53706, USA

**Keywords:** newborn screening (NBS), severe combined immunodeficiency (SCID), premature newborns, gestational age, T-cell receptor excision circles (TRECs)

## Abstract

The Wisconsin Newborn Screening (NBS) Program began screening for severe combined immunodeficiency (SCID) in 2008, using real-time PCR to quantitate T-cell receptor excision circles (TRECs) in DNA isolated from dried blood NBS specimens. Prompted by the observation that there were disproportionately more screening-positive cases in premature infants, we performed a study to assess whether there is a difference in TRECs between full-term and preterm newborns. Based on de-identified SCID data from 1 January to 30 June 2008, we evaluated the TRECs from 2510 preterm newborns (gestational age, 23–36 weeks) whose specimens were collected ≤72 h after birth. The TRECs from 5020 full-term newborns were included as controls. The relationship between TRECs and gestational age in weeks was estimated using linear regression analysis. The estimated increase in TRECs for every additional week of gestation is 9.60%. The 95% confidence interval is 8.95% to 10.25% (*p* ≤ 0.0001). Our data suggest that TRECs increase at a steady rate as gestational age increases. These results provide rationale for Wisconsin’s existing premature infant screening procedure of recommending repeat NBS following an SCID screening positive in a premature infant instead of the flow cytometry confirmatory testing for SCID screening positives in full-term infants.

## 1. Introduction

Severe combined immunodeficiency (SCID) is a fatal genetic disorder that causes T-cell lymphopenia, leaving the patient without a functioning immune system. Newborns with SCID appear healthy at birth but are extremely vulnerable to infection; early diagnosis and treatment with curative approaches such as bone marrow transplants are key to successful intervention [1]. In 2004, SCID was recognized as a disorder that met important criteria for inclusion in newborn screening (NBS) at a Centers for Disease Control and Preventions conference [2]. In 2005, Chan and Puck published a seminal paper describing the measurement of T-cell receptor excision circles isolated from NBS dried blood spots as a potential population-based approach to identifying infants with SCID [3]. In 2008, the Wisconsin NBS program was the first to implement a molecular screening assay for SCID using optimized DNA isolation techniques and RT-PCR for TRECs [4,5,6,7,8]. Between 2008 and 2010, three additional states began NBS SCID screening: Massachusetts in 2009 and California and New York in 2010 [9,10,11,12]. In 2010, SCID screening was added to the Recommended Uniform Screening Panel (RUSP) [13]. Over the next 8 years, states and territories of the US added SCID to their panels, and by the end of 2018, all US NBS programs were screening for SCID [9]. During this same time, population-based NBS for SCID also moved forward in other countries. Early international work was conducted in Japan and Sweden, and programs in the UK, Israel, Spain, the Netherlands, Brazil, Taiwan, France, Saudi Arabia, Canada, and Norway have all published on their SCID NBS experiences [14,15,16,17,18,19,20,21,22,23,24,25,26,27,28,29].

TRECs are generated during T-cell maturation in the thymus, and, therefore, TRECs directly reflect thymic output and the maturity of the T-cells [30]. In 2013, Ward et al. reported that premature infants posed a challenge to population-based screening due to their immature immune systems [31]. In the last few years, several NBS programs have published their experience in screening the premature infant population for SCID and the difficulties of balancing false-positive results with ensuring screening catches true SCID cases in the premature infant population [32,33,34,35,36]. Here, we report our study results assessing the relationship between TREC copy numbers and gestational age in weeks. These results are informative and worthy of been taken into the consideration when managing SCID results for the premature infant population.

## 2. Materials and Methods

Between January 1, and 30 June 2008, every preterm infant with a specimen collection time ≤ 72 h after birth with a screening-negative result (≥25 TRECs/µL) and no missing data was included in our study (*N* = 2510). In addition, two full-term newborns with similar collection times and screening-negative results (≥25 TRECs/µL) from the same assay plates were also included (*N* = 5020). Every individual was coded with a sequential ID number for analysis and associated plate number, gestational age, TREC count, and birth weight. DNA was isolated using the protocol outlined in Baker et al. 2009, and real-time quantitative PCR for measuring TRECs was subsequently performed [4]. The 7530 sample results were obtained from 490 separate 96-well plates. The TREC data were transformed to the log scale before analysis to obtain constant variability across all gestational weeks.

For the linear regression analysis of TREC copy number versus gestation age, to account for the variability in plate processing, we modeled the effect of plate as a random effect in a mixed effects model. We tested the straight-line mixed effects model against a model that allowed for a “broken line”. The broken line fit was not a significant improvement over the straight-line model (*p* = 0.9100). We conclude that a single straight-line fit both preterm and full-term gestation data. The relationship between the TREC copy number and gestational age was then estimated using separate univariate linear regression with an added blocking factor to adjust for the effect of the experimental plate. There was no evidence of lack of fit in the residual plots.

## 3. Results

Aggregate population-level TREC copy numbers in five gestational age group categories are shown in Figure 1. Using the log transformed TREC copy number data, a one-way ANOVA shows statistically significant differences across the means of these groups (*p* value < 0.0001).

Figure 2 shows that in our linear regression analysis, the TREC copy number in newborns increases at a steady rate of 9.8% per week as gestational age increases. The estimated increase in the TREC copy number for every additional week of gestation is 9.76% increase per week with a 95% confidence interval of 9.21% to 10.31% and a *p*-value less than 0.0001. The *y*-axis of the plot is a log-scale axis. Note that the increments in adjusted TREC are not equal. In this way, the axis points are labeled in the original units, but the data are log transformed so that the linear fit can be seen. The *x*-axis values are “jittered” so that the data points do not all line up on top of each other. This is carried out to give a sense of the distribution of the adjusted TREC values for the weeks where there are large amounts of data. There are much larger sample sizes for the later gestational age groups than for the very premature gestational age categories.

Using the raw data again, three weight categories are shown in Figure 3: a very-low-birth-weight group (≤1500 g), a low-birth-weight group (1500–2500 g), and a normal-birth-weight group (>2500 g). Table 1 shows the pairwise comparison of these weight categories using a one-way ANOVA of the log transformed TREC values. The statistical difference between categories was significant. TRECs in very-low and low-birth-weight newborns are significantly lower than normal birth weight newborns.

## 4. Discussion

Preterm newborns pose special challenges to newborn screening. They are generally developmentally immature and need special clinical interventions, which can interfere with the newborn screening process. For many NBS assays, the interpretation of results for premature neonates is achieved using special algorithms or reference ranges to try to minimize the impact of these confounders. With the data we present above, we show that SCID NBS poses the same challenges, with prematurity directly impacting the TREC copy number. Using the same recommendations for premature infants as for full-term infants following SCID screening-positive results could lead to unnecessary flow cytometry confirmatory testing.

Wisconsin NBS has been using a split follow-up algorithm for SCID screening that considers gestational ages. The same TREC cutoff value is applied to all infants screened. For full-term newborns with an out-of-range TREC result, or for any newborn with TREC values of zero on the newborn screen, flow cytometry confirmatory testing is recommended [37]. For newborns with an out-of-range TREC result with a gestational age < 37 weeks, a repeat NBS is requested at 14 days of age, 30 days of age, and then monthly until 3 months of age or when the infant is discharged from the NICU. These recommendations are stated on the NBS reports issued by the program, and primary care providers are contacted if specimens are not collected within the recommended timeframe. When an infant’s age plus gestational age reaches 37 weeks, follow-up recommendations for full-term newborns would be applicable to this infant. Additional NBS collections can place burden on NICUs and NBS systems, so programs must consider the timing and frequency of repeat collections. Prior to SCID screening implementation, a similar practice existed in Wisconsin NBS for congenital hypothyroidism based on birth weights, so there was no need to collect additional specimens for repeating SCID screening. Most screening-positive infants have screening-negative results on subsequent repeat screens. For example, in 2020, out of 59,896 screened infants, there were 23 premature infants with out-of-range SCID screening results, and all had negative results in subsequent repeated screening.

## Figures and Tables

**Figure 1 IJNS-07-00040-f001:**
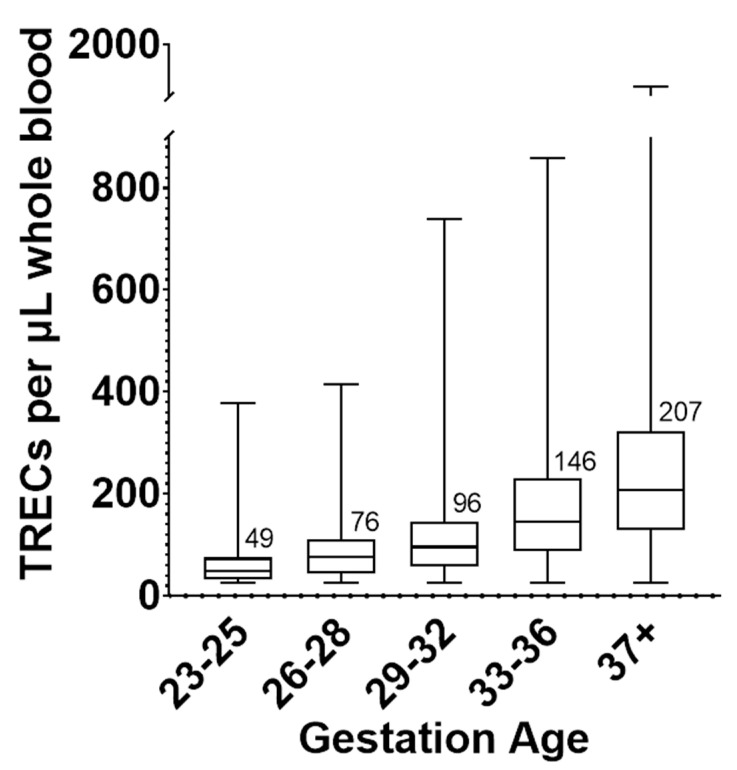
Box and whisker plot of TREC copy number by five gestational age (GA) categories. For 23–25 GA, *N* = 33; 26–28 GA, *N* = 54; 29–32 GA, *N* = 259; 33–36 GA, *N* = 2164; 37 + GA, *N* = 5020. The line within each box and the numbers printed next to the box represent the median of each group. The top and bottom of each box are the 25 and 75 percentiles, respectively. The whiskers represent the range of TREC values.

**Figure 2 IJNS-07-00040-f002:**
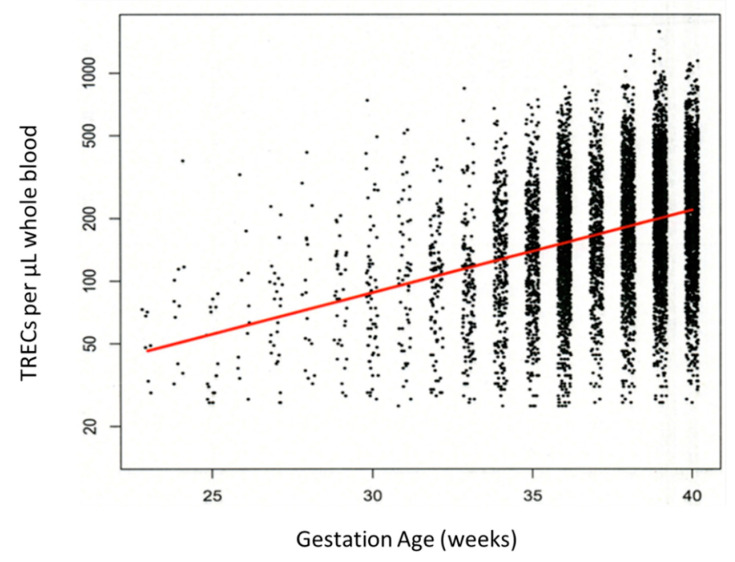
Plot of gestational age versus TRECs. The *y*-axis of the plot is in the log scale so that the linear regression line can be seen (red line). TREC copy numbers increase 9.6% per additional week of gestation with 95% CI of 8.95% to 10.25% (*p* < 0.0001).

**Figure 3 IJNS-07-00040-f003:**
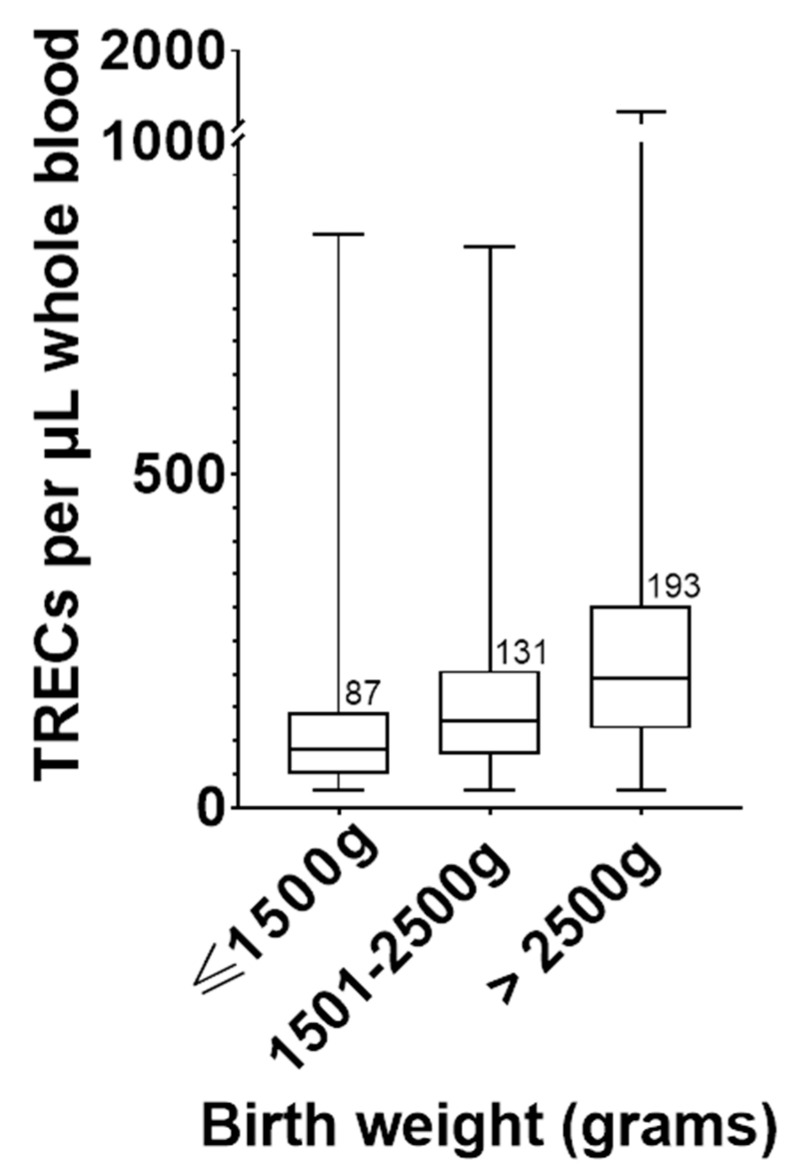
Box and whisker plot for TREC copy number by birth weight category. For the ≤1500 g category, *N* = 220; 1500–2500 g category, *N* = 1068 samples; >2500 g category, *N* = 6242. The line within each box and the numbers printed next to the box represent the median of each group. The top and bottom of each box are the 25 and 75 percentiles, respectively. The whiskers represent the range of TREC values.

**Table 1 IJNS-07-00040-t001:** One-way ANOVA of the log transformed TREC values for three weight groups shown in Figure 3, using multiple comparisons. Data were corrected for multiple comparisons using Šídák’s multiple comparisons test.

Weight Categories	Mean Diff.	95.00% CI of Diff.	Adjusted *p* Value
1501–2500 g vs. ≤1500 g	0.1451	0.09519 to 0.1949	<0.0001
>2500 g vs. 1501–2500 g	0.1668	0.1445 to 0.1891	<0.0001

## Data Availability

The data are not publicly posted, and can be requested by contacting the corresponding author.

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
