# Peer review of "Newborn Screening for Severe Combined Immunodeficiency: Do Preterm Infants Require Special Consideration?"

_2409-515X, 2021, doi:10.3390/ijns7030040_

Round 1

Reviewer 1 Report

This manuscript compares T-cell receptor excision circles (TREC)-levels between full-term and preterm newborns in a de-identified cohort from 2008. The data confirms previous observations of lower TREC-levels in preterms and justifies the adjustment of the screening algorithm for premature newborns with TREC-levels outside the normal range. The paper is concise, clearly written and logical. The use of linear regression analysis to observe a ‘steady state increase’ in TRECs is an innovative aspect as other studies mainly use correlation analysis. However, low TREC-levels have been previously described in larger cohorts and the novelty of the paper is somewhat limited. The authors could have made an extra effort in comparing screening strategies for preterms between different NBS programs. 

Some remarks:

  1. M&M section: why have the authors chosen to exclude newborns with abnormal TREC screens? Does this not result in selection bias leading to higher median TREC values in the preterm population? As you are excluding more screen positive cases from your preterm population compared to the full-term population…
  1. How was screen negative defined? TRECs below a certain cut-off?

  2. The authors matched preterms and full term controls based on collection time. Was matching also performed for other confounding factors such as gender or receiving blood transfusion?
  1. Figure 1: Could the authors add group size (N=) per gestational age category in the figure?
  1. The text states that TREC-copy numbers were statistically lower between the groups in Figure 1. What statistical test was used for group comparison and could you add levels of significance to the figure or mention a P-value for Figure 1?

  2. The study from Rechavi et al. 2017 suggests a leap in TREC increase between 28 and 30 weeks gestation. Did the authors observe similar results and would they agree with the theory that moderate to late preterms with screening positive results should follow the regular screening algorithm for full term newborns (for example as recently described by the Polish-German pilot study, Gizewska et al. 2020).
  1. Figure 3: Could the author also include group size (N=) in Figure 3 (which is mentioned in the text)?
  1. Figure 3: Why have the author chosen for a different way of depicting data in Figure 3? I think it is preferable to use box-plots similar to Figure 1.
  1. Table 1: Could you specify the test used for pairwise comparison? ANOVA?
  1. Is repeating the NBS card at different ages part of the NBS program or are pediatricians already contacted at this stage? Could you maybe briefly describe how parents are informed during this process?
  1.  It would be nice to reflect on other NBS programs and their chosen screening strategies for preterms. Some programs only repeat TREC analysis at 34 weeks, others at 37 weeks. Other programs use different cut-off values for preterms. Could the authors comment on these choices (pro’s/con’s) and provide arguments for their recommended screening strategy to help other NBS programs? 

One minor comment

  • Use of the abbreviation NBS should be consequent throughout the manuscript

Reviewer 2 Report

The manuscript by Anne E. Atkins et al evaluated the differences in TREC values between preterm and full-term infants. The data are impactful for clinical immunologists and neonatologists who are often faced with the challenges of interpreting and acting upon a low TREC value in an NBS-SCID report for a preterm infant. The manuscript is well written and the data are presented well. I have the following comments:

1. Figure 1: Label the x-axis (gestational age in weeks).

2. It is interesting to note that even in very low birth weight infants (<1500 grams), the mean TREC copy number/mL was still reasonably greater than the cut-off for an abnormal result. While it is important to repeat TREC analysis in preterm infants until they reach term birthweight or age, can the authors comment on these results further? From Figure 2, how often did preterm infants in the very low birth weight category have abnormal TREC results that were clinically concerning based on the Wisconsin criteria?

3. The authors should note that a TREC value of zero in a preterm infant is still significant and should be followed with an immune evaluation. The challenge here is that there is not a lot of data on robust ranges for lymphocyte subsets in these ages. The following reference for lymphocyte subset ranges can be included in the discussion: Amatuni GS et al. Reference intervals for lymphocyte subsets in preterm and term neonates without immune defects. J Allergy Clin Immunol. 2019 Dec;144(6):1674-1683.
